**www.cambridge.org/qrd**

# Reduced protein solubility – cause or consequence in amyloid disease?

Max Lindberg[1] , Jing Hu[2], Emma Sparr[2] and Sara Linse[1]

[1]Biochemistry and Structural Biology, Lund University, Lund, Sweden and [2]Division for Physical Chemistry, Lund University, Lund, Sweden

## Perspective

Alzheimer associated amyloids; amyloid complexes; biomolecular systems; protein biophysics; physical chemistry; chaperones

**Corresponding author:**
Sara Linse;
Email: sara.linse@biochemistry.lu.se

### Abstract

In this perspective, we ask the question whether the apparently lower solubility of specific proteins in amyloid disease is a cause or consequence of the protein deposition seen in such diseases. We focus on Alzheimer's disease and start by reviewing the experimental evidence of disease-associated reduction in the measured concentration of amyloid β peptide, Aβ42, in cerebrospinal fluid. We propose a series of possible physicochemical explanations for these observations. These include a reduced solubility, a reduced apparent solubility, as well as a long-lived metastable state manifested in healthy individuals as a free concentration of Aβ42 in the solution phase above the solubility limit. For each scenario, we discuss whether it is most likely a cause or a consequence of the observed protein deposition in the disease.

## Introduction

### Protein precipitation in amyloid diseases

Neurodegenerative and metabolic diseases are associated with deposition of specific proteins, typically only one or a few proteins per disease. Well-known examples are Alzheimer's disease (AD), Parkinson's disease, Huntington's disease, and type II diabetes (Chiti and Dobson, 2006, 2017; Buxbaum et al., 2022). The protein aggregates are elongated fibrils of high aspect ratio called amyloid fibrils and the deposits may be termed plaques, tangles, Lewy bodies, and so forth. Although the deposits are typically dominated by one amyloid protein, they may contain, for example, lipids, length variants of the amyloid protein as well as other proteins such as, for example, chaperones (Gellermann et al., 2005; Brinkmalm et al., 2019). A key question regarding all those diseases is whether precipitation occurs due to reduced protein solubility upon changes in the peptide or its environment, or maybe before emergence of the disease the protein was in a metastable state for long time (Portugal Barron and Guo, 2023; Guo, 2021; Roos et al., 2021; Durrant 2024)? This is linked to the overarching question of whether the observed reduction in the apparent solubility is a cause or consequence of protein precipitation in amyloid disease.

### Amyloid β peptide in ADs

In this perspective, we will start by discussing the situation in AD, which is associated with the extracellular precipitation of amyloid β peptide, Aβ, and intracellular precipitation of the protein tau (Scheltens et al., 2016; Bondi et al., 2017; Lane et al., 2018; Walsh and Selkoe, 2020; Scheltens et al., 2021). Although the clinical symptoms of the disease were described in the early 1900s, the two proteins, tau and Aβ, were identified some 40–50 years ago (Weingarten et al., 1975; Glenner and Wong, 1984). According to the amyloid cascade hypothesis, based on clinical observations, the formation of Aβ plaques precedes the formation of tau tangles (Hardy and Higgins, 1992; Selkoe and Hardy, 2016), and the phosphorylation of tau seems to be an intermediate event triggering its aggregation (Oliveira et al. 2017). Recent studies have identified key differences in the expression levels of specific proteins in neurons, notably chaperons as well as proteins involved in reelin signaling and the clearance system, which correlate with cell survival rate in AD (Mathys et al., 2023, 2024).

In both healthy and disease conditions, Aβ of various lengths are constantly produced from the amyloid β precursor protein, APP, through the action of proteases called β- and γ-secretases (Tagawa et al., 1991). Although the precision of the β-secretase, also called BACE1, which cleaves the N-terminal end of Aβ, is relatively high, but with some variation (Kaneko et al., 2014; Welzel et al., 2014), the γ-secretase is less precise in action leading to several C-terminal length variants. Of these, Aβ40 is the most common and Aβ42 is the most strongly associated with AD (McGowan et al., 2005).

A general observation in AD seems to be a drop in the concentration of at least one alloform of Aβ compared to healthy individuals. Specifically, the 42-residue alloform, Aβ42, has been reported to decrease about twofold in cerebrospinal fluid in AD compared to healthy controls

CAMBRIDGE
UNIVERSITY PRESS

(Motter et al., 1995; Galasko et al., 1998; Andreasen et al., 1999; Portelius et al., 2006; Portelius et al., 2010; Agarwal et al., 2012; Willemse et al., 2018). Such changes can be observed several years before the emergence of the clinical symptoms and used as a diagnostic criterion (Andreasen et al., 1999; Blennow and Hampel, 2003; Buchhave et al., 2012; Willemse et al., 2018).

## *Possible molecular reasons?*

In this perspective, we will discuss some possible molecular explanations for the observed decrease in Aβ42 concentration in AD versus healthy controls. We will put emphasis on the possible physicochemical explanations, which often rely on changes in the composition of the biological system. There might be a handful of possible physical explanations that follow a small number of mechanisms. We will cover metastability versus stability of the system. We will discuss the possibilities for reduced solubility due to changes in peptide physical properties leading to decreased repulsion or increased attraction, and how changes in alloform composition may change the state of the system as a whole. We will discuss apparent solubility versus solubility, including solubilization by small molecules and molecular chaperones.

## *Metastability versus stability – was it in a metastable state before AD?*

Amyloid formation is a phase transition with the solution phase containing monomers in solution, and some small fraction of small aggregates, oligomers, and the solid phase containing monomers in fibrils. At equilibrium, the chemical potential of monomers in both phases is equal (Eq. 1):

$$\mu_m = \mu_m^o + RT\ln([m]) = \mu_f = \mu_f^o \qquad (1)$$

where $\mu_m$ and $\mu_m^o$ are the chemical and standard chemical potentials, respectively, of free monomers in the solution phase, that is, $\mu_m$ is dependent on their concentration, $[m]$. In contrast, $\mu_f$, the chemical potential of monomers in fibrils is concentration independent and equal to the standard chemical potential, $\mu_f^o$, of monomers in fibrils. The difference in stability of the fibrils and monomers ($\mu_f^o - \mu_m^o$) under the particular solution conditions thus sets the level of the solubility of the peptide, s, that is, the highest concentration of free peptide in solution that can exist at equilibrium (Eq. 2) in that environment.

$$s = \exp\left\{\frac{\mu_f^o - \mu_m^o}{RT}\right\} \qquad (2)$$

If the system is not yet in equilibrium, but in a metastable state, the concentration of the free monomer is above the solubility limit (blue line in Figure 1). In this range, the system is unstable, and may collapse with the formation of fibrils and reduction of the free monomer concentration down to the solubility value. As time goes by, the width of the metastable zone decreases and at infinite time it is totally gone (red line in Figure 1). A possible explanation for the observed drop in Aβ42 concentration in AD versus healthy may thus be that there is no change in solubility, but the peptide is indeed kept in a metastable state for very long time, over decades. This is in contrast to cases of covalently modified peptide, in which case the standard chemical potential of monomers in solution and in fibrils may be differently affected, leading to changes in solubility. This will be discussed below. When aggregates have nucleated, new aggregates may form at close to an exponential rate and the peptide concentration in solution will drop to the equilibrium solubility (Figure 1A). The chemical potential of the free peptide depends only on the total concentration of free peptide, so it will not matter that there is continuous peptide turnover through anabolism and catabolism, production and clearance, as long as the concentration remains constant.

In this scenario (Figure 1a), there is indeed no change in monomer solubility between the healthy state and AD, it is just a matter of time until nucleation of aggregates leads to escape of the metastable state. In support of this scenario, several investigators report the findings of decreased levels of Aβ42 in solution after introduction of seed aggregates (Kane et al., 2000; Meyer-Luehmann et al., 2006; Langer et al., 2011; Roos et al., 2021).

## *Changes in the balance of anabolism and catabolism*

A second possibility is a disruption of the balance between anabolism and catabolism, which may lead to changes in peptide concentration and may put the system at risk (Agarwal et al., 2006). In this second scenario, there may be age-related changes in Aβ production, or the catabolic pathways may also start to fail with

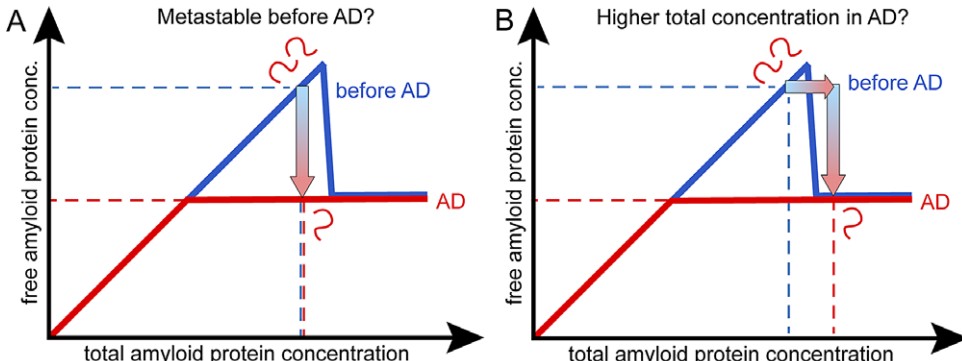

**Figure 1. The protein is metastable before AD**. At equilibrium, for a phase transition, the free and total concentrations of the amyloid protein are equal up to the solubility limit, above which the free monomer concentration remains constant at the solubility level irrespective of the total monomer concentration. (**A**) In one scenario, this is the situation in AD (red line), the higher free concentration observed before AD (blue line) is due to the protein remaining in a metastable state for very long time. (**B**) In another scenario, the amyloid protein is in a metastable state before AD, but in this case, the system escapes the metastable zone because of an increase in the total concentration of the amyloid protein in AD. The gradient arrows indicate the direction of change from healthy to AD. Dashed lines indicate possible values for the free and total concentration before AD (blue) and in AD (red), compatible with these two scenarios.

age, leading to reduced clearance, and increased risk for precipitation. There are reports on clearance during night sleep of aggregates formed during the day (Chauhan et al., 2017). If the clearance system starts to fail, this may lead to accumulation of precipitates and collapse of the metastable zone, because the total concentration goes above the metastable zone (Figure 1B).

In this scenario, there is again no change in solubility of the amyloid protein between the healthy state and AD; it is just a matter of increased total concentration, which leads to escape of the metastable state. In support of this scenario are reports of altered anabolism or catabolism in AD leading to higher Aβ42 levels (Saido, 1998; Iwata et al., 2005; Ullah et al., 2021). A striking example is the situation in individuals with Down's syndrome, for which chromosome 21 trisomy leads to 50% higher Aβ42 production without a concomitant increase in catabolism and early onset AD (Podlisny et al., 1987).

### Covalent modifications – was it in a more soluble form before AD?

A third possible explanation for the observed drop in Aβ42 concentration in AD versus healthy is that the solubility of the peptide is lower in AD due to changes in the peptide covalent structure. If the standard chemical potential of monomers in solution and in fibrils are differently affected, this leads to a change in solubility (Eq. 2). Age-dependent covalent modifications of the peptide may disrupt the system through reduced repulsive or increased attractive intermolecular interactions. Due to covalent changes, the amyloid protein may thus be present below the solubility limit of the unmodified peptide in the healthy state, but above the solubility limit of the modified peptide in AD. Some age-related covalent modifications, such as oxidation (Head et al., 2001), pyroglutamate formation (Saido et al., 1995), and N-terminal truncation (Bayer and Wirths, 2014) display limited reversibility and may lead to reduced solubility of Aβ42. The onset of peptide precipitation and system escape from the metastable state may be triggered by such covalent modifications and may be understood as reduced peptide–peptide repulsion/increased peptide–peptide attraction. The peptide may therefore be stable at the higher concentration before AD than after, because before it was in its regular form and after in a less soluble form (Figure 2A).

Alternatively, the peptide may be stable at a higher concentration before AD than after, because before it was in a modified and more soluble form in the healthy state and in its regular less soluble form in AD (Figure 2B). Phosphorylation is a more rapid and reversible covalent modification, observed at positions 8 and 26, which may on the contrary lead to increased solubility of Aβ42 due to increased electrostatic repulsion (Milton, 2001; Arnés et al., 2020; Kuzin et al., 2022; Sanagavarapu et al., 2024).

In this scenario, there is a change in solubility of the amyloid protein between the healthy state and AD, governed by covalent modification of the peptide before AD or in AD.

Genetic variations leading to point mutations in the Aβ peptide is a more permanent modification and regardless of the mutations making the peptide more or less soluble, it has its own phase diagram, different from the wild-type peptide, and the mutant peptide may be covered by any of the other scenarios above and below.

In the case of tau, phosphorylation at multiple sites, that is, hyper-phosphorylation, seems to drive its aggregation (De Felice et al., 2008). Tau is a positively charged protein, and phosphorylation adds negative charge, reducing the net charge of tau, thereby reducing the electrostatic repulsion and favoring aggregation. One may thus argue that the reduced solubility of phosphorylated compared to non-phosphorylated tau may be a cause of AD pathology. However, according to the amyloid cascade hypothesis, this follows after Aβ aggregation, so the reduced solubility of tau may also be seen as a consequence of early events that lead to the pathology.

### Modulation of the alloform ratios

Age-related changes in the activities and specificities of the β- and γ-secretases that produce Aβ from the amyloid precursor protein, APP, may lead to modulation of the Aβ alloforms ratio. The number of hydrophobic residues at the C-terminus varies due to the more or less imprecise cleavage of APP in the membrane by the γ-secretase. Studies of whole brain extracts (Brinkmalm et al., 2019) and individual plaques (Beretta et al., 2024) have revealed a great variation in alloform ratios between individuals and between the center and periphery of plaques. The N-terminal cleavage by the β-secretase is of higher precision, but also here significant length variation has been observed with both truncations and extension of the N-terminus (Bayer and Wirths, 2014; Kaneko et al., 2014; Welzel et al., 2014). Although the aggregation propensity varies among alloforms, as well as their ability to form co-fibrils with Aβ42, some alloforms may increase the solubility of Aβ42 or delay its precipitation (Cukalevski et al., 2015, Braun et al., 2022). Thus, a

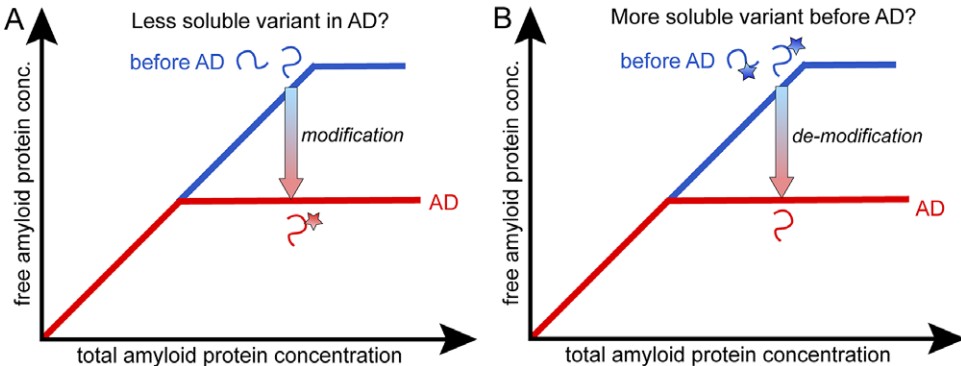

**Figure 2. Covalent modifications of the amyloid protein changes its solubility**. In cases of covalent modifications, the phase transition and solubility may shift to lower total monomer concentration in AD. (**A**) This may be due to the amyloid protein being is in its regular form in the healthy state before AD, and covalently modified to a less soluble form in AD. (**B**) Alternatively, the amyloid protein may be covalently modified to a more soluble form in the healthy state but present in its regular less soluble form in AD. The gradient arrows indicate the direction of change from healthy to AD.

change in the alloform composition may increase the risk of amyloid precipitation.

## Changes in the solution conditions

The observed drop in Aβ42 concentration in AD versus healthy may be due to the solubility of the peptide being lower in AD compared to healthy because of changes in the solution composition of the amyloid protein environment. This may lead to modulation of the relative importance of different types of attractive and repulsive intermolecular interactions. Changes in pH between compartments may affect electrostatic interactions via titration of protein charges and lead to changes in both long- and short-range interactions. Changes in ionic strength may affect electrostatic interactions through altered screening effects and lead to changes in both long- and short-range interactions. Changes in the composition of salt ions and other ionic substances between the extracellular and intracellular environments or between compartments may modulate the strength of hydrophobic interactions and lead to changes in short-range interactions. Changes in temperature, although in a small range in living humans, change the solubility of amyloid protein through modulation of hydrophobic interactions. Even if some of the changes listed in this paragraph may be small in a living system, the effect on a system close to the solubility limit, or in a metastable state, may still be detrimental.

## Solubility versus solubilization

It is also possible that the observed drop in Aβ42 concentration in AD versus healthy may be due to the apparent solubility of the peptide being lower in AD due to solubilization of the amyloid protein by chaperones or small detergent-like molecules. In this scenario, the amyloid protein concentration in the solution phase will be higher before AD than during AD, not because the protein is more soluble but because it is solubilized in co-assembled micelle-like aggregates (Figure 3A). In this scenario, the emergence of AD would be related to a drop in concentration of the detergent-like molecules with age. In this scenario, there is no change in solubility of the amyloid protein between the healthy state and AD, but the amyloid protein is better solubilized before AD to sustain a higher concentration in solution.

Examples of micelle-forming lipids with detergency properties are lysolipids, fatty acids, gangliosides, and sphingosides. All these lipids have in common that they have higher water solubility compared to typical membrane lipids. The free lipid monomers may also interact with Aβ monomers or smaller protein oligomers, and may thereby reduce amyloid formation and precipitation. There are several studies reporting on alteration in brain ganglioside content with age, for example, observations of decreasing contents of gangliosides in general or of specific gangliosides such as GM1 (Segler-Stahl et al., 1983; Svennerholm et al., 1989; Kracun et al., 1992; Sipione et al., 2020). Changes of ganglioside contents in patients with amyloid diseases, including Parkinson's disease, Huntington's disease, and AD were also observed (Sipione et al., 2020). The role of gangliosides in amyloid disease is, however, yet not elucidated. Some studies showed the co-assembly of amyloid beta proteins with micelles of GM1 (Hu et al., 2023), lyso-GM1 (Utsumi et al., 2009), and sodium dodecyl sulfate (Shao et al., 1999). The formation of such micelle–amyloid protein co-assemblies can alter the apparent solubility of the amyloid protein, and possibly influence the neurotoxicity of the amyloid protein.

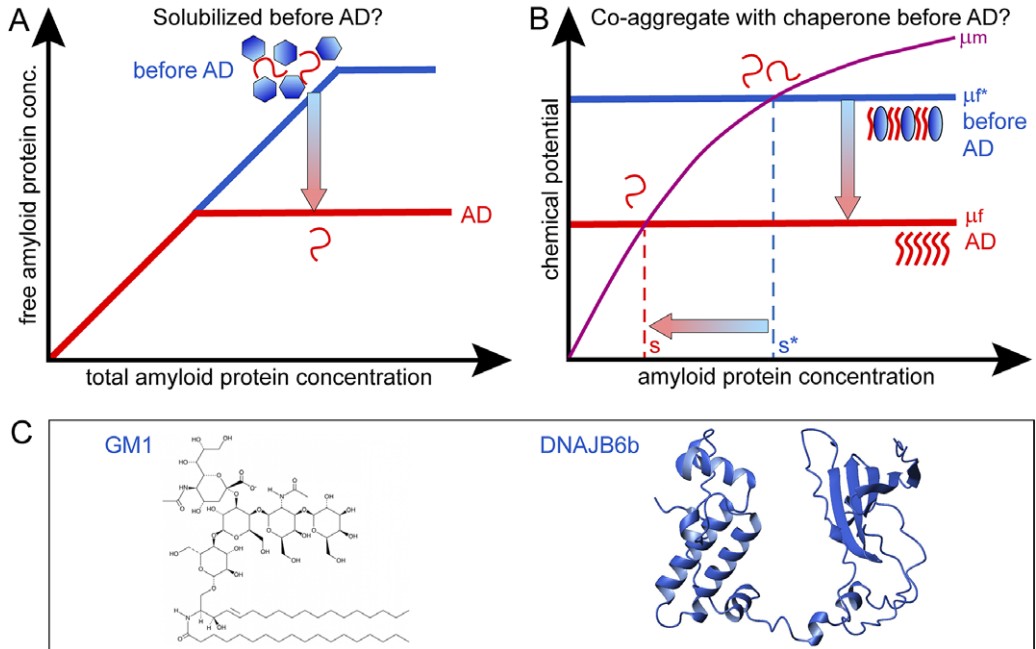

**Figure 3. The amyloid protein is in a co-aggregate with other molecules in solution or in the solid phase**. (**A**) In one scenario, the protein is solubilized by chaperones or detergent-like molecules before AD such that the monomer concentration in the solution phase is higher before AD than during AD. (**B**) In another scenario, the monomer in solution is always free but in the solid phase, it may be in a co-aggregate with a chaperone or detergent-like molecule, with enough different properties that the chemical potential of the monomer is higher in the co-aggregate compared to pure amyloid fibrils. This means that the monomer concentration in solution can be sustained at higher concentration in the presence of the co-assemblies compared to in the presence of the pure fibril. The gradient arrows indicate the direction of change from healthy to AD. $\mu_m$ (purple curve), $\mu_f$ (red line), and $\mu_{f*}$ (blue line) are the chemical potential of amyloid monomers in solution, pure amyloid fibrils, and amyloid-chaperone co-aggregates, respectively. The solubilities of the amyloid peptide in the absence and presence of chaperone are denoted s (dashed red line) and s* (dashed blue line), respectively. (**C**) Example of solubilizing molecules in the form of the ganglioside lipid GM1 (with one more hydrocarbon chain than the related lyso-GM1) and the chaperone DNAJB6b.

## Co-aggregates with chaperones in the solid phase

It is also possible that a higher amyloid protein concentration in solution is sustained in the healthy state because in the solid phase there are co-aggregates between the amyloid protein, in which the chemical potential of amyloid monomers, $\mu_{f*}$, is higher compared to monomers in pure fibrils, $\mu_f$ (Linse et al., 2021). In this scenario, as the chaperone system starts to fail with age, the emergence of pure (or more pure) amyloid fibrils with lower chemical potential will drive the monomer concentration in solution to lower values to retain the requirement for equal chemical potential of the monomers in both phases.

In this scenario, there is indeed a change in solubility of the amyloid protein between the healthy state and AD, brought about by a variant solid phase consisting of chaperone-amyloid co-aggregates. In support of this scenario are reports of reduced chaperone gene expression in older (73 ± 4 years of age) compared to younger (36 ± 4 years of age) individuals (Brehme et al., 2014), and of increased solubility of Aβ42 in the presence of the chaperone DNAJB6b (Månsson et al., 2018).

## Cause or consequence?

As discussed, the observed lower concentration of the Aβ42 peptide in AD compared to the healthy state may or may not be a sign of lower solubility of the peptide in AD. We therefore need to change the initial question and rather ask: is the observed reduction of Aβ42 concentration in solution in AD a cause or consequence of the disease? Moreover, we here ask for possible physicochemical explanations of the observation rather than the primary biological/pathological event. For each possibility discussed above, we thus rather ask whether it is likely a cause or a consequence of the observed protein deposition in the disease.

In the first scenario of the healthy state being a metastable state, there is a change in monomer concentration in solution, but not in solubility, between the healthy state and AD. It is just a matter of time and nucleation of aggregates leading to escape of the metastable state. This scenario is thus a possible cause of protein deposition in the disease, and it is a matter of probability, that is, good or bad luck, whether an individual manages to stay in the metastable state throughout life or develops the disease. An intriguing question regarding this scenario is what are the properties of the system that allows the metastable zone to be so long-lived at a width that covers up to about two times the solubility. Or is it rather a matter of exactly this fact, that the total concentration is indeed only about twice the solubility that allows the system to remain metastable for so long?

In the second scenario, there is again no change in solubility of the amyloid protein between the healthy state and AD; the escape of the metastable state is a matter of increased total concentration. This scenario could be a cause of protein deposition in the disease, with over-production of the peptide or decreased catabolism being the triggering event. However, this scenario could also be a consequence; if the system regulates free peptide in solution rather than total peptide, there is no regulatory mechanism to limit the total concentration once the solubility limit is reached; any extra production would add to the solid phase rather than the solution phase, meaning a continuous increase in the deposited amount.

Covalent modifications may indeed lead to a change in solubility of the amyloid protein between the healthy state and AD. A change in peptide modification is likely a consequence of what happens in the environment of the peptide in AD via altered activities of the modifying enzymes. However, the associated drop in solubility may

be a triggering event that becomes a cause of the protein deposition in the disease. If the change in monomer concentration in solution is a result of covalent modifications of the amyloid protein, this may thus be both a cause and a consequence of protein deposition.

Changes in alloform ratios are a consequence of changes in secretase specificity but may be a cause of the disease due to altered co-aggregation and co-catalysis between alloforms when their molar ratio changes.

A change in solution conditions may lead to a change in solubility such that the new solubility lies below the total concentration. In such case, the change in solution conditions may be a cause of protein deposition. Even if the changes in solution conditions may be small in a living system, the effect on a system that was close to its solubility limit may still be detrimental.

The amyloid protein may be better solubilized before AD to sustain a higher concentration in solution. A decrease in peptide solubilization may then be a consequence of what happens in the environment of the peptide in terms of altered concentrations of the molecules with detergency properties. Is the associated drop in apparent solubility also a triggering event that becomes a cause of the protein deposition in the disease? Impaired solubilization may indeed be both a cause and a consequence of protein deposition.

In the last scenario covered in this perspective, there is indeed a change in solubility of the amyloid protein between the healthy state and AD, brought about by a variant solid phase consisting of chaperone-amyloid co-aggregates. A failure in the chaperone system as we age may thus be a cause of decreased protein monomer concentration in solution and of protein deposition in the disease.

## Outlook

The current perspective presents a set of possible scenarios that may explain the observations of lower Aβ42 concentration in solution phase in AD compared to healthy individuals. We discuss for each of these suggestions whether it is most likely a cause or a consequence of amyloid protein deposition in AD. Attempts to validate of falsify these scenarios, for example, using isotope-enriched Aβ42 to measure its solubility in body fluids from AD and healthy individuals, may lead to new important insights toward the pathology of AD.

**Open peer review.** To view the open peer review materials for this article, please visit http://doi.org/10.1017/qrd.2024.12.

**Data availability statement.** This perspective contains no new data.

**Author contribution.** All authors contributed to this perspective.

**Financial support.** This work was supported by the Swedish Research Foundation Vetenskapsrådet (VR, grant number 2015-00143 to SL and 2019-02397 to ES), the European Research Council (ERC, grant number 101097824 to SL), and the Knut and Alice Wallenberg Foundation (KAW, grant number 2022.0059 to SL).

**Competing interest.** None.

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
