## [Reviewer Report]

Reduced protein solubility - cause or consequence in amyloid disease?

Lindberg and colleagues wrote a Perspective to highlight the current knowledge of biochemistry and biophysics of proteins that are involved in Alzheimer’s disease and other neurodegenerative diseases. They focus on the aspect of protein stability and aggregation, both Abeta and tau. They address the “Chicken and egg” question in molecular terms, namely, what is the root cause of Alzheimer’s disease from protein stability and aggregation that have been observed for over a century in Alzheimer’s disease. They also asked why it takes very long time, most often until old age of >70 or older to have the unset of the disease. The other exceptions in some isolated populations that younger people also develop the Alzheimer’s disease from the genetic mutations, however, some of them are more resilient because other factors, such as high expression of Reelin in the brain. https://en.wikipedia.org/wiki/Reelin

Minor points:

1) Figure 2A, it is suggested that they label the blue/pink arrow with a word “modification”. Such words/terms are often used above the arrows in chemical and enzymatic reactions. They could also provide an example as tau phosphorylation in the figure legend.

2) Figure 2B, it is suggested that they label the blue/pink arrow with a word “de-modification”. They could also provide an example as Abeta Serine-8 phosphorylated forms (pSer8-Aβ) before and after de-phosphorylation in the figure legend.

3) Figure 3A, for the cartoon molecules in blue, it is suggested to provide examples of detergents such as GM1, lyso-GM1, or chaperon and add in the figure legend.

4) Figure 3B, for the cartoon molecules in blue, it is suggested to provide example, such as DNAJB6b. Please write in the legend what is blue *s and red s; blue µf* and red µf, even they are mentioned in the main text. Clarity is very important to retain readers’ attention.

5) This reviewer suggests them to cite a recent paper in Nature about the Reelin effect the delays the onset of Alzheimer’s disease.

Mathys H, Boix CA, Akay LA, Xia Z, Davila-Velderrain J, et al. Single-cell multiregion dissection of Alzheimer’s disease. Nature. 2024 Aug;632(8026):858-868. doi: 10.1038/s41586-024-07606-7. Epub 2024 Jul 24. PMID: 39048816.

Mathys H, Peng Z, Boix CA, Victor MB, Leary N, et al. Single-cell atlas reveals correlates of high cognitive function, dementia, and resilience to Alzheimer’s disease pathology. Cell. 2023 Sep 28;186(20):4365-4385.e27. doi: 10.1016/j.cell.2023.08.039. PMID: 37774677; PMCID: PMC10601493.

Lopera, F., Marino, C., Chandrahas, A.S. et al. Resilience to autosomal dominant Alzheimer’s disease in a Reelin-COLBOS heterozygous man. Nature Medicine 29, 1243–1252 (2023). https://doi.org/10.1038/s41591-023-02318-3

After making these minor changes, this reviewer highly recommends publication as a Perspective in QRB Discovery.

---

## [Editor Report]

Dear Sara,

Could you please revise your ms in the light of the single reviewer’s comments. We are expecting some additional comments from a second reviewer who however is positive. Therefore, to give you time for the minor revision I decide to invite you do it now.

---

## [Reviewer Report]

The authors have revised the manuscript and added the suggested references. The final manuscript is greatly improved. This reviewer highly recommends publication in QRB Discovery without further delay.

---

## [Reviewer Report]

Authors have now revised (minor revision) their manuscript which is now in good order for publication. Accept!